# Let-7b-5p sensitizes breast cancer cells to doxorubicin through Aurora Kinase B

**Murat Kaya[1], Asmaa Abuaisha[2], Ilknur Suer[3], Selman Emiroglu[4,5], Semen Önder[6], Evren Onay Ucar[7], Mustafa Nuri Yenerel[8], Sukru Palanduz[1], Kivanc Cefle[1], Sukru Ozturk[1], Zeyneb Kurt ORCID[9]***

1 Istanbul Faculty of Medicine, Department of Internal Medicine, Division of Medical Genetics, Istanbul University, Capa, Fatih, Istanbul, Turkey, 2 Research Center, Biruni University, Istanbul, Turkey, 3 Istanbul Faculty of Medicine, Department of Medical Genetics, Istanbul University, Capa, Fatih, Istanbul, Turkey, 4 Istanbul Faculty of Medicine, Department of General Surgery, Division of Breast Surgery, Istanbul University, Capa, Fatih, Istanbul, Turkey, 5 Department of General Surgery, Biruni University School of Medicine, Biruni University, Istanbul, Turkey, 6 Istanbul Faculty of Medicine, Department of Pathology Capa, Istanbul University, Fatih Istanbul, Turkey, 7 Faculty of Science, Department Of Molecular Biology and Genetics, Istanbul University, Capa, Fatih, Istanbul, Turkey, 8 Istanbul Faculty of Medicine, Department of Internal Medicine, Division of Hematology, Istanbul University, Capa, Fatih, İstanbul, Turkey, 9 Information School, The Wave, The University of Sheffield, Sheffield, United Kingdom

* z.kurt@sheffield.ac.uk

**Data Availability Statement:** All human breast cancer-related microRNA Microarray data files are available from the GEO database, www.ncbi.nlm. nih.gov/geo, (accession numbers: GSE45666,

## Abstract

MicroRNAs (miRNAs) are small, non-coding RNAs that regulate the expression level of the target genes in the cell. Breast cancer is responsible for the majority of cancer-related deaths among women globally. It has been proven that deregulated miRNAs may play an essential role in the progression of breast cancer. It has been shown in many cancers, including breast cancer, that aberrant expression of miRNAs may be associated with drug resistance. This study investigated the effect of let-7b-5p, detected by bioinformatics methods, on Dox resistance through the Aurora Kinase B (*AURKB*) gene. In silico analysis using publicly available miRNA expression, GEO datasets revealed that let-7b-5p significantly downregulated in BC. Further in silico studies revealed that of the genes among the potential targets of let-7b-5p, *AURKB* was the most negatively correlated and may be closely associated with Dox resistance. Expression analysis via quantitative PCR confirmed that let-7b-5p was downregulated and *AURKB* was upregulated in breast cancer tissue samples. Later, functional studies conducted with MCF-10A, MCF-7, and MDA-MB-231 cell lines demonstrated that let-7b-5p inhibits cancer cells through *AURKB* and sensitizes them to Dox resistance. In conclusion, it has been shown that the let-7b-5p/*AURKB* axis may be significant in breast cancer progression and the disruption in this axis may contribute to the trigger of Dox resistance.

## Introduction

Breast cancer (BC) is a prevalent malignancy resulting in mortality and morbidity among women globally [1,2]. Recurrence and metastases, mostly resulting from BC's resistance to

GSE154255, GSE38167). All human breast cancer-related mRNA datasets were collected from TCGA-BRCA (https://portal.gdc.cancer.gov/projects/TCGA-BRCA) and from the GEO database, www.ncbi.nlm.nih.gov/geo (accession numbers: GSE100925 and GSE229571).

**Funding:** This research was supported by the Council of Higher Education Research Universities Support Program project, ADEP-Istanbul University, TSA-2023-39483. The funders had no role in study design, data collection and analysis, decision to publish, or preparation of the manuscript.

**Competing interests:** I have read the journal's policy and the authors of this manuscript have the following competing interests: -The corresponding author, Dr Zeyneb Kurt, is a member of the editorial board for the PLOS ONE journal. -The rest of the authors have declared that no other competing interests exist.

conventional therapy, are the main causes of mortality among those affected with BC [3]. Many individuals receiving chemotherapy experience a poor initial response or acquired resistance to chemotherapy over time, known as multidrug resistance (MDR) [3,4]. The molecular mechanisms underlying drug resistance in cancers, including BC, are still not fully elucidated. Various alterations in components of pathways for signaling or critical regulators of cell death execution can inhibit the effect of drugs [5]. Understanding the reasons for drug resistance is therefore vital and clinically significant in improving the outcome of BC patients.

MicroRNAs (miRNAs) are approximately 18–24 nucleotides long and have essential functions in post-transcriptional gene regulation [6]. Dysregulation of miRNAs involved in fundamental biological activities such as cell division, proliferation, and apoptosis has been associated with hundreds of diseases, including BC [2,7,8]. Uncovering the relationships between miRNAs and their target genes may be necessary in elucidating disease mechanisms and developing new treatment strategies. Studies show that miRNAs may play roles in various drug resistance mechanisms [9,10]. For example, silencing miR-21, which is overexpressed in BC, has been demonstrated to enhance tamoxifen or fulvestrant sensitivity by inhibiting the *PI3K/AKT/mTOR* pathway in ER-positive BC cells [11]. Safaei et al. demonstrated that miR-200c increases Doxorubicin (Dox) sensitivity through downregulating *MDR1* gene expression in BC cells [12]. Let-7b-5p, a let-7 family member, is one of the important tumor suppressor miRNAs and regulates the cell cycle. Abnormal reduction in let-7b-5p expression levels has been linked to various cancers, including BC [13–16]. For example, it has been showed that let-7b-5p has a tumor suppressor role in osteosarcoma by blocking *IGF1R*. Aurora kinase B (*AURKB*) is a chromosomal passenger protein complex component that regulates cell cycle progression. AURKB deregulation has been detected in various cancers, and its overexpression is commonly associated with tumor cell invasion, metastasis, and drug resistance [17]. However, there is no information about the link between let-7b-5p/*AURKB* axis and BC. In the present work, analysis of BC-related miRNAs and genes was carried out using the bioinformatics approaches. The expression levels of the chosen miRNA (let-7b-5p) and its potential target gene (*AURKB*) were validated in BC tissue samples, and the Dox resistance link between the let-7b-5p/*AURKB* axis was explored using BC cell lines.

## Material and methods

### Bioinformatics methods utilized for the identification of differentially expressed miRNAs (DEmiRs)

Human breast cancer-related miRNAs were derived from microarray geo datasets GSE45666 (101 tumor samples, 15 adjacent tissue samples), GSE154255 (10 tumor samples, 10 adjacent tissue samples), and GSE38167 (31 tumor samples, 23 adjacent tissue samples) using GEO database (www.ncbi.nlm.nih.gov/geo). Differentially expressed miRNAs (DEmiRNAs) whose expression changed between tumor tissue samples and the adjacent tissues in the datasets were identified with GEO2R (http://www.ncbi.nlm.nih.gov/geo/geo2r) and the parameters were specified to be p < 0.05 and log-fold change (logFC) > 1. The differential expression has been defined as a change in the logFC. A logFC >1 indicates that the miRNAs have changed at least twice (fold change >2.0), suggesting significant differences across groups.

### Filtering of miRNAs with prognostic significance in BC

Overlapping DEmiRs between the datasets GSE45666, GSE154255, and GSE38167 were identified. Using KM-plotter (Kmplot) [18], one of the most advanced online survival analysis tools, it was investigated whether the detected miRNAs significantly affected overall survival

(OS) of BC. In Kmplot tool, The Cancer Genome Atlas (TCGA) and Molecular Taxonomy of Breast Cancer International Consortium (METABRIC) data can be analyzed separately for OS analysis. Kmplot was used to determine the BC-OS relationship of overlapping miRNAs utilizing both TCGA and METABRIC data for further filtering. Separation of patients into high-low-risk level groups has been performed by automated identification of the best cutoff value, which is the default setting in the Kmplot user interface. All potential values among the lower and upper quartiles are explored and the best-performing cutoff, which concludes with the most significant p-value or the highest hazard ratio is selected automatically by the tool.

## Determination of the most suitable miRNA for wet lab study

Overlapping miRNAs associated with BC-OS were searched in Pubmed with the keywords "miRNA name, breast cancer". The miRNA with the highest results score was selected as the most suitable miRNA for the wet lab experiments.

## Identifying differentially expressed mRNAs (DEMs)

Gene expression differences between tumor and normal tissues were analyzed through Gene Expression Profiling Interactive Analysis 2 (GEPIA2) [19] portal using TCGA-BC data. Genes that met the logFC>2, p<0.001 criteria were considered as DEMs. At the same time, BC-related RNA-seq gene expression datasets GSE100925 (36 tumor and 36 adjacent tissue samples), GSE229571 (6 tumor and 6 adjacent tissue samples) were obtained from the GEO database, and DEMs between tumor tissues and normal tissues were detected using the GEO2R tool (genes with LogFC > 2, p < 0.001 were taken into account).

## Identifying matches between overlapping miRNA and genes

Possible target genes of the selected miRNA were examined using the miRNet tool [20]. Overlapping genes were detected between the TCGA-BC, GSE100925, GSE229571 datasets, and the potential targets of let-7b-5p reported by miRNet.

## KEGG, GO, and PPI analysis of overlapping genes

Enrichr [21] tool was utilized to carry out Gene Ontology (GO) and Kyoto Encyclopedia of Genes and Genomes (KEGG) pathway analysis on overlapping DEMs. p < 0.05 was selected as the significance level. The protein-protein interaction (PPI) network was created using the Search Tool for the Retrieval of Interacting Genes (STRING) [22] database. The cut-off level was set to a degree > 10.

## Correlation analysis of selected miRNA and overlapping genes

The Pearson correlation scores between selected miRNA and overlapping genes were calculated with CancermiRNome [23].

## Cancer stage-wise analysis of selected gene

ExplORRNet is a web-based interactive tool [24] for investigating stage-specific miRNA expression patterns and their interactions with mRNA and lncRNA in individuals with breast and gynecological malignancies. ExplORRNet was used to compare selected gene expressions between different stages of BC.

## In vitro investigation of selected miRNA and gene

**Patient specimens.** Between November 2020 and November 2022, we acquired 72 pairs of human BC samples consisting of tumors and adjacent normal tissues from cases in which breast surgery was performed at the Department of General Surgery, Istanbul Faculty of Medicine Hospital, Istanbul University (Istanbul, Turkey). All patients provided their written informed consent. (Ethics number: 29624016–050.99–903).

## Cell lines and culture conditions

MCF-7 and MDA-MB-231 BC cells were gifted by Prof. Dr. Mehmet Topcul Istanbul University, Faculty of Sciences, Department of Biology, and MCF-10A (normal breast epithelial cells) were provided by Prof. Dr. Neslihan Abaci from Istanbul University Aziz Sancar Institute. The MCF-7/Doxorubicin resistance (Dox) and MDA-MB-231/Dox Dox drug-resistant variants of the MCF-7 and MDA-MB-231 cell lines, respectively, were established by stepwise selection at Dox concentrations ranging from 0.5 to 30 μmol/L of parental cells for five months [25].

All cells were cultured in DMEM (EcoTech Biotechnology, Erzurum, Turkey) with 10% FBS (Gibco, Grand Island, NY, USA) and 1% penicillin/streptomycin (Sigma, St. Louis, MO, USA) in a conditioned incubator at 37°C with 5% $CO2$.

## MiRNA mimic transfection and barasertib treatment

All of the cells (parental MCF-7, MDA-MB-231 cells and dox/MCF-7, MDA-MB-231 cells) were seeded at 60% confluence into 96- or 6-well culture plates. After 24 hours, using the supplier's method for transient 30 nM let-7b-5p mimic or miR-NC (Thermo Fisher Scientific, Waltham, MA, USA), cells were transfected via lipofectamine 3000 reagent (Invitrogen, Carlsbad, CA, USA). AURKB inhibitor Barasertib (Thermo Fisher) dissolved in dimethyl sulfoxide (DMSO). The transfected or treated cells were evaluated after 24, 48, or 72 hours of culture.

## qRT-PCR analysis

The relative expression of chosen genes were evaluated using quantitative real-time PCR (qRT-PCR). Total RNA was isolated from matched 72 BC tissue specimens or BC cell lines via TRIzol reagent (Invitrogen, San Diego, CA). A NanoDrop spectrophotometer was used to assess the quality and quantity of RNA samples. To investigate mRNA and miRNA expression levels, equal amounts of RNA from the samples were reverse transcribed into cDNA using a cDNA reverse transcription kit (Thermo Fisher) and TaqMan Kit (Invitrogen), respectively. The qRT-PCR experiments were carried out to detect mRNA expression using 5x HOT FIRE qPCR Mix Plus (Solis-Biodyne, Tartu, Estonia). *GAPDH* expression was utilized to normalize gene expression. As an internal control, *GAPDH* was employed for the gene and RNU43 for the miRNA. Each reaction was repeated at least twice. Relative expression was assessed using the $2^{-\Delta\Delta Ct}$ method. Gene sensitivity and specificity were assessed using the receiver operating characteristic (ROC) curve for BC tissue specimens' qRT-PCR results.

## Detection of chemosensitivity

After 24 hours of cultivation in 96-well plates, transfected cells were treated with Dox doses ranging from 0.01 to 120 μM for 48 hours. Cell viability was then assessed using the Cell Counting Kit-8 (EcoTech Biotechnology). The Dox half maximum inhibitory concentration ($IC_{50}$) values were computed using GraphPad Prism.

## Migration assay

The migration capacities of MCF-7/ Dox, MCF-7, MDA-MB-231/ Dox, and MDA-MB-231 cells were assessed using a wound healing experiment. All cells were seeded into each well of a six-well plate and cultivated to 90% confluency. Scraping the plate with a 100 μl pipette tip created a wound in the center of the cell monolayer. Images of the migrated area were captured with an optical microscope at 0 and 48 hours after let-7b-5p mimic transfection and/or *AURKB* inhibitor treatment. Images of the migrating cells were captured after 48 hours under an inverted microscope.

## Western blot analysis

Cells were washed twice with ice-cold PBS after 48 h transfection and collected for further investigation. RIPA protein extraction buffer was used to extract of total proteins (Thermo Fisher). Protein concentration was measured with the SMART TM BCA Protein Assay Kit. Protein samples (25 μg/well) were separated by SDS-PAGE and transferred to PVDF membranes with the Bio-Rad wet tank transfer system. Following 1 hour of blocking with 5% non-fat dry milk (in TBST) at room temperature, the membranes were incubated overnight with primer antibody at 4°C. After washing with TBST, they were incubated with the secondary antibody for 1 hour at room temperature. Following TBST washing, the membranes were examined with a SuperSignal TM West Pico PLUS Chemiluminescent Substrate kit. Anti-GAPDH loading control monoclonal antibody was used for the normalization of data. Image-Lab 6.1 software (Bio-Rad Bio-Rad Laboratories, Inc., Hercules, CA, USA) was used to measure band density.

## Apoptosis assay

The Annexin V/PI assay was used to evaluate the cells' apoptosis using the Annexin V-FITC Apoptosis Detection kit (Beckman Coulter). Briefly, cells were obtained using 0.25% trypsin-EDTA, washed twice with cold PBS, and resuspended in a cold binding buffer. Cells were treated for 15 minutes with 3 μl of Annexin V-FITC and 5 μl of propidium-iodide. Beckman Coulter Navios flow cytometer (Navios 3L10) was used for apoptosis examination. 400 micro-liters of annexin binding buffer was added to the ready cells and read in the Kaluza analysis program. The apoptosis rates (early–late) and necrosis were determined. The results were analyzed and recorded using Kaluza software.

## Statistical analysis

The bioinformatics data utilized in the study were gathered from public databases and in silico tools. GraphPad Prism 10 software was used for data analysis and visualization of statistical charts. The Student's t-test was employed to assess group differences. Values are provided as mean ± standard deviation, with P-values as $^*P < 0.05$, $^{**}P < 0.01$, $^{***}P < 0.001$ indicating statistical significance.

## Results

### Three BC datasets revealed 19 shared dysregulated miRNAs

We identified 145 miRNAs in the GSE45666, 108 miRNAs in the GSE154255, and 395 miR-NAs in the GSE38167 as differentially expressed in tumor vs. normal tissue samples. Among them, 19 differentially expressed miRNAs were shared between the datasets (Table 1 and Fig 1A).

**Table 1. Shared DEmiRNAs in GSE45666, GSE154255 and GSE38167 GEO datasets.**

| MiRNAs | GSE45666 | | GSE154255 | | GSE38167 | |
|---|---|---|---|---|---|---|
| | P.Value | logFC | P.Value | logFC | P.Value | logFC |
| miR-199b-5p | 5.55e-07 | -2.55 | 2.55e-01 | -1.70 | 9.14e-06 | -2.30 |
| miR-210 | 3.06e-09 | 2.33 | 6.31e-02 | 4.72 | 9.39e-05 | 2.26 |
| miR-145-3p | 3.29e-10 | -2.31 | 1.58e-01 | -3.13 | 1.93e-06 | -2.05 |
| let-7c-5p | 4.03e-09 | -1.90 | 6.30e-03 | -1.69 | 1.59e-05 | -1.91 |
| miR-125b-2-3p | 1.58e-11 | -2.60 | 1.58e-01 | -3.95 | 3.78e-06 | -1.87 |
| miR-542-5p | 2.41e-06 | -1.73 | 1.71e-01 | -1.58 | 1.06e-05 | -1.79 |
| miR-429 | 4.82e-04 | 1.45 | 1.68e-01 | 3.29 | 9.63e-04 | 1.78 |
| miR-142-3p | 1.21e-07 | 2.37 | 9.54e-02 | 4.35 | 1.84e-04 | 1.74 |
| miR-132-3p | 1.25e-08 | -1.63 | 5.58e-01 | -1.85 | 1.06e-04 | -1.74 |
| miR-139-3p | 2.83e-05 | -1.69 | 1.83e-01 | -2.03 | 2.47e-03 | -1.63 |
| miR-486-5p | 1.82e-08 | -2.13 | 7.11e-03 | -6.55 | 2.73e-04 | -1.56 |
| miR-10b-3p | 1.05e-10 | -2.10 | 4.17e-01 | -2.94 | 1.10e-02 | -1.50 |
| miR-1290-5p | 1.11e-07 | 2.50 | 4.70e-01 | 1.47 | 8.15e-03 | 1.26 |
| let-7b-5p | 2.74e-07 | -1.40 | 1.13e-02 | -1.49 | 4.41e-04 | -1.11 |
| miR-513a-5p | 3.69e-04 | -1.22 | 5.30e-01 | -1.00 | 6.92e-02 | -1.12 |
| miR-30a-3p | 4.44e-08 | -2.49 | 1.41e-01 | -2.16 | 6.65e-02 | -1.04 |
| miR-126-3p | 1.79e-09 | -2.37 | 2.16e-01 | -1.31 | 9.81e-03 | -1.03 |
| miR-572 | 1.08e-06 | -1.70 | 4.47e-02 | -2.66 | 2.77e-02 | -1.01 |
| miR-484 | 2.13e-03 | 1.02 | 8.30e-02 | 1.43 | 3.09e-03 | 1.00 |

## Let-7b-5p, miR-1290-5p and let-7c-5p were related to a poor prognosis of BC

The effect of overlapping miRNAs on BC-OS was examined using TCGA and METABRIC clinical data. According to the OS analysis results, only three of the 19 overlapping miRNAs were shown to influence BC-OS significantly within both databases (**Fig 2**, **Table 2**). An investigation of the relationship between these three miRNAs and BC was conducted in PubMed, and the results showed that these miRNAs were closely connected with BC. As a result, all three miRNAs might be good candidates for future in vitro investigations. Among these three strong candidate miRNAs, let-7b-5p was chosen for further investigation in vitro for the purpose of this research.

## 14 potential target genes of let-7b-5p with elevated expression, including AURKB, might be essential in BC

Potential target genes of selected let-7b-5p were investigated using the miRNet tool. "Human, miRBase ID, miRTarbase" options are selected in miRNet to detect target genes. Among all, 1215 potential target genes of let-7b-5p were listed. Then, overlapping genes were searched between BC datasets and these 1215 genes. In conclusion, 14 overlapping genes were determined among the dysregulated genes between the GSE100925, GSE229571, TCGA-BC, and let-7b-5p potential targets in miRNet tool (**Fig 1B and 1C**, **and Table 3**). The increased expression of these genes within all three datasets might be a key sign of BC.

## Overlapping genes may be linked to numerous essential transcription factors

In silico analysis results showed that the 14 overlapping genes may be associated with many important transcription factors including *TP53* and *MYC*, *BRCA1*, *BRCA2*, *FOXO1* etc.

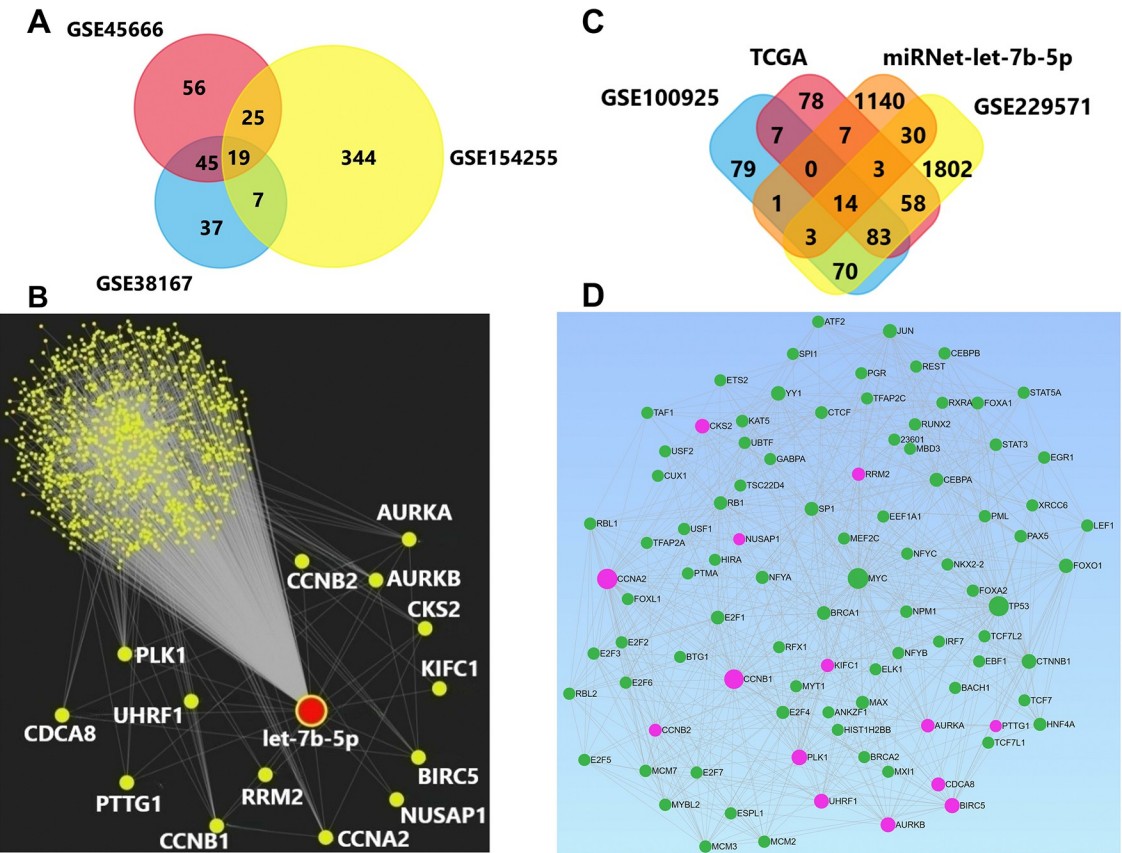

**Fig 1.** (A) DEmiRNAs shared between the three datasets (B) let-7b-5p had the potential to target a total of 1215 genes, including 14 overlapping genes, according to the miRNet tool. C) Venn diagram of overlapping genes between TCGA-BC data, GSE100925, GSE229571 geo datasets and miRNet-let-7b-5p potential targets. (D) To identify the potential transcription factors associated with overlapping 14 genes, we employed official human gene names, RegNetwork, and P<0.05 criteria in the miRNet tool. The results revealed that there were 80 transcription factors (with 467 edge connections). Nodes in pink: Overlapping genes, nodes on green: Transcription factors.

(**Fig 1D**). The results suggest that because alterations to the expression of 14 genes can affect the expression of many transcription factors, further research into the roles of these genes in BC processes may be critical in understanding the molecular mechanisms underlying disease initiation and development.

## Overlapping genes may play important roles in critical processes

The findings of STRING analysis, which analyzed the interaction between overlapping genes, revealed that the interaction between 14 genes was more than expected (p<1.0e-16) (**Fig 3A**). According to the results of the KEGG research, overlapping genes may have roles in several critical pathways, including the *p53* and *FoxO* signaling pathways (**Fig 3B**).

## Findings show that the *AURKB* gene is a stronger candidate for let-7b-5p

In silico research revealed that let-7b-5p can target *AURKB* from the 3'UTR region (**Fig 3C**). Further more it was confirmed using a luciferase reporter assay [26,27]. Using TCGA-BC data, the correlation between let-7b-5p and 14 possible target genes was explored. The *AURKB* gene was shown to have the highest significant negative correlation among the overlapping genes

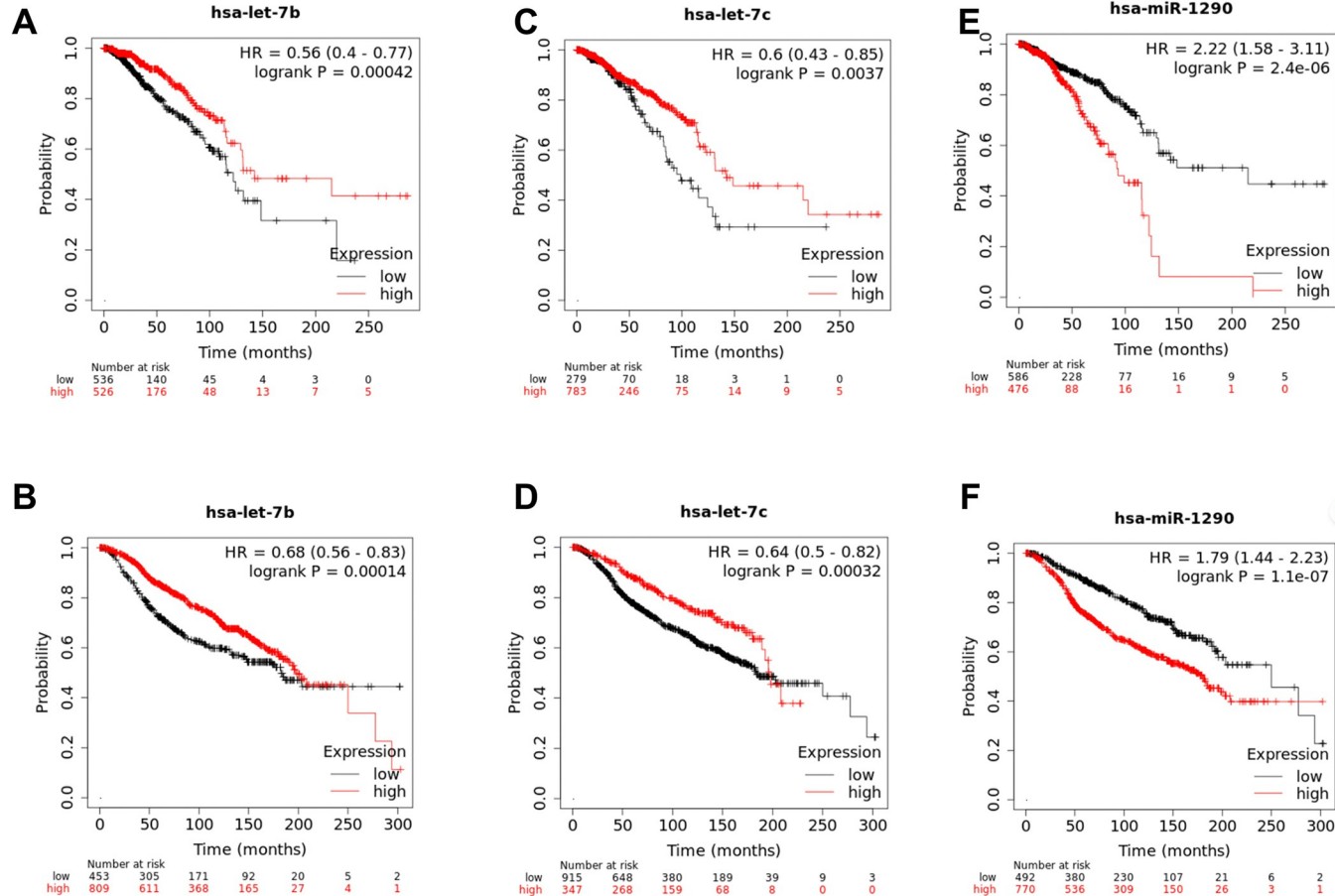

**Fig 2. miRNAs that may have prognostic significance based on both TCGA-BC and METABRIC datasets.** The diminished expression of let-7b-5p (A-B) and let-7c-5p (C-D) miRNAs has a negative effect on BC-OS. The increased expression of miR-1290-5p (E-F) also had a harmful impact on BC-OS. Figures (A), (C), and (E) were produced using TCGA-BC data. Figures (B), (D), and (F) were created based on METABRIC data.

for let-7b-5p (**Fig 3D**). Furthermore, the literature research revealed that let-7b-5p/*AURKB* is associated with drug resistance in some malignancies [26,27]. These in silico and literature data suggest that the let-7b-5p/*AURKB* axis should be further studied in BC.

### *AURKB* may be an important gene for BC

According to the stage analysis results based on TCGA-BC data, *AURKB* was shown to be highly expressed in all BC stages. Furthermore, it is noteworthy that *AURKB* gene expression rises as the stage level increases (**Fig 3E**)**.** The correlation analysis between the *AURKB* gene and other overlapping genes demonstrated that all overlapping genes had a strong positive correlation with *AURKB* (**Table 4**). These findings suggest that the let-7b-5p/*AURKB* interaction should be studied further in BC.

### In vitro analysis results showed that let-7b-5p and AURKB had opposite expression patterns in BC

Let-7b-5p and *AURKB* expression levels in patient tumor tissue and adjacent normal tissue samples were investigated by qRT-PCR. The clinicopathological characteristics of the patients are demonstrated in **Table 5**. The expression level of let-7b-5p was significantly reduced in

**Table 2. Investigation of whether the DEmiRNAs shared between the GSE45666-GSE154255-GSE38167 GEO datasets have an impact on the BC-OS, comparatively in TCGA and METABRIC datasets.** (OS analysis was employed via KMplot tool).

| Shared DEmiRNAs | TCGA (n: 1078) | METABRIC (n: 1262) |
|---|---|---|
| hsa-miR-199b-5p | Not Found | Yes |
| hsa-miR-210 | No | Yes |
| hsa-miR-145-3p | Not Found | Not Found |
| hsa-let-7c | Yes | Yes |
| hsa-miR-125b-2-3p | Not Found | Not Found |
| hsa-miR-542-5p | Not Found | Yes |
| hsa-miR-429 | Yes | No |
| hsa-miR-142-3p | Not Found | No |
| hsa-miR-132-3p | Not Found | Not Found |
| hsa-miR-139-3p | Not Found | Yes |
| hsa-miR-486-5p | Not Found | No |
| hsa-miR-10b-3p | Not Found | Not Found |
| hsa-miR-1290 | Yes | Yes |
| hsa-let-7b | Yes | Yes |
| hsa-miR-513a-5p | Not Found | Yes |
| hsa-miR-30a-3p | Not Found | Not Found |
| hsa-miR-126-3p | Not Found | Not Found |
| hsa-miR-572 | Yes | No |
| hsa-miR-484 | Yes | No |

n: The number of the patients.

cancer tissue samples (P < 0.001). On the contrary, *AURKB* expression was significantly increased in BC tissue samples (P < 0.001) (**Fig 4A and 4B**). These results were consistent with our in silico analysis results. We also investigated the let-7b-5p and *AURKB* expression levels in MCF-10A, MCF-7, MCF-7/Dox, MDA-MB-231, and MDA-MB-231/Dox cells using qRT-PCR. Let-7b-5p expression was significantly reduced in all parental BC cancer cells and

**Table 3. Overlapping let-7b-5p potential target genes in TCGA-BC, GSE100925 and GSE229571 datasets.**

| Genes | TCGA-BC | | GSE100925 | | GSE229571 | |
|---|---|---|---|---|---|---|
| | P. Value | logFC | P. Value | logFC | P. Value | logFC |
| *RRM2* | 4.62e-189 | 3.76 | 7.84E-18 | 3.40 | 2.63e-19 | 5.06 |
| *BIRC5* | 1.26e-186 | 3.40 | 1.53E-13 | 3.24 | 7.75e-17 | 4.79 |
| *PTTG1* | 3.08e-208 | 3.25 | 3.72E-17 | 2.78 | 2.22e-10 | 3.56 |
| *NUSAP1* | 1.58e-197 | 3.20 | 1.52E-17 | 2.66 | 2.48e-18 | 3.56 |
| *CCNB2* | 5.52e-185 | 3.11 | 1.36E-17 | 2.94 | 1.69e-07 | 2.60 |
| *CKS2* | 7.87e-239 | 3.05 | 2.28E-17 | 2.55 | 1.33e-09 | 2.98 |
| *UHRF1* | 2.56e-203 | 3.00 | 1.77E-12 | 2.24 | 1.61e-18 | 3.34 |
| *CCNB1* | 9.81e-209 | 2.98 | 2.02E-18 | 2.45 | 1.16e-10 | 2.76 |
| *AURKA* | 7.11e-174 | 2.77 | 2.23E-18 | 2.84 | 3.47e-07 | 3.17 |
| *KIFC1* | 3.59e-170 | 2.74 | 6.50E-13 | 2.21 | 9.29e-07 | 2.52 |
| *AURKB* | 5.92e-145 | 2.67 | 2.17E-16 | 2.96 | 9.61e-09 | 3.78 |
| *PLK1* | 7.23e-158 | 2.64 | 7.54E-19 | 3.19 | 1.40e-10 | 2.88 |
| *CDCA8* | 2.04e-164 | 2.45 | 2.44E-18 | 2.73 | 1.13e-13 | 3.13 |
| *CCNA2* | 2.48e-130 | 2.09 | 7.27E-15 | 2.48 | 5.39e-12 | 2.95 |

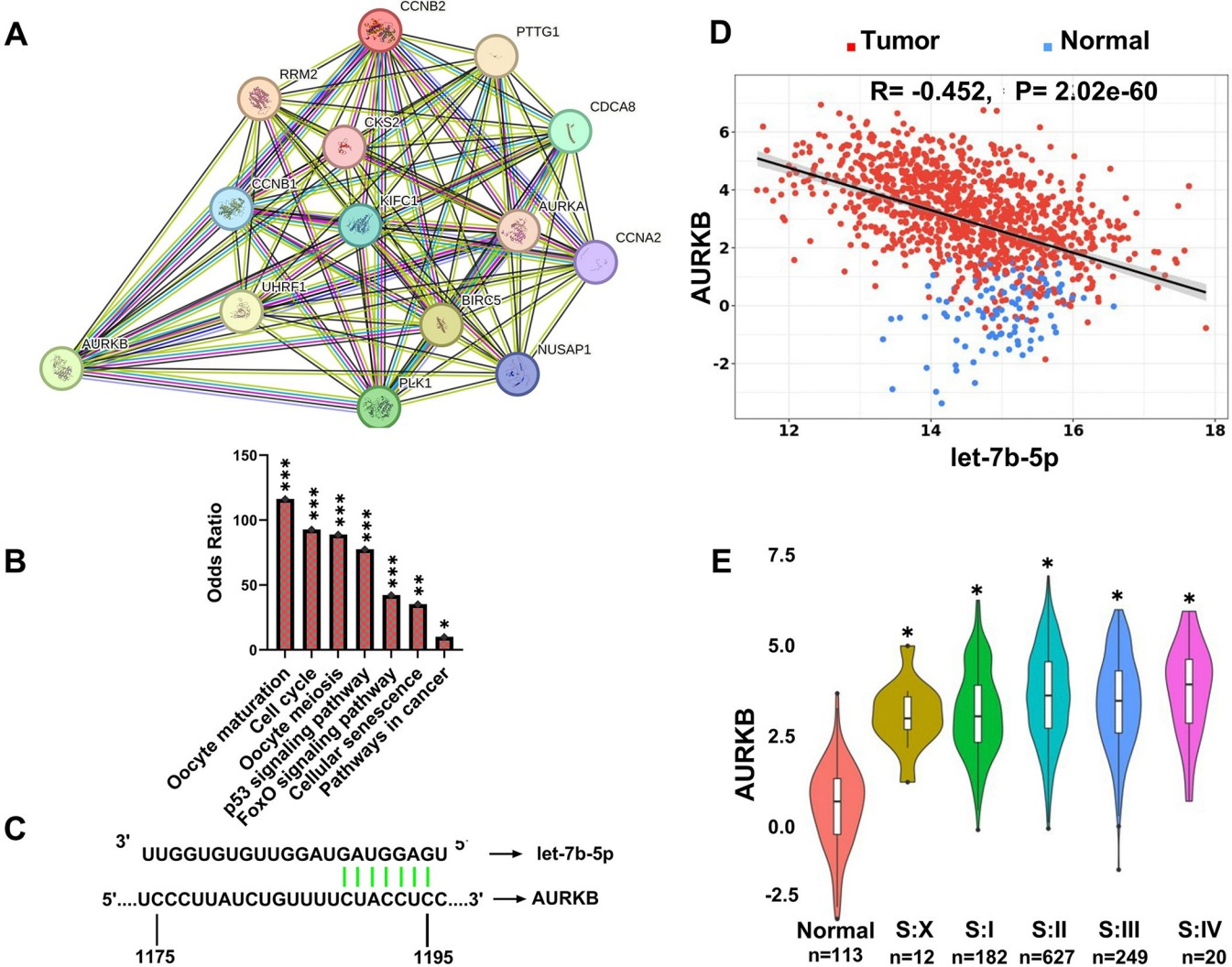

**Fig 3.** (A) STRING Analysis results: Nodes: 14, edges: 89, average node degree: 12.7, avg. local clustering coefficient: 0.98, the expected edges: 8, PPI p-value: < 1.0e-16 (B) Enrichment analysis was performed with 14 official gene names using Enrichr. According to the KEGG 2021 pathway analysis results, it was determined that 14 genes may play a role in important pathways, such as the oocyte maturation p53 signal pathway. (C) Schematic depiction of AURKB, showing the probable let-7b-5p target site. (D) According to TCGA-BC data, the Pearson correlation between let-7b-5p and AURKB expressions was significant (investigated using the CancermiRNome tool). (E) According to TCGA-BC data, AURKB expression was found to be significantly increased in all stages compared to normal tissue (analysis was performed using ExplORRnet).

Dox-resistant BC cells compared to MCF-10A cells (P < 0.05), while *AURKB* expression was increased (P < 0.05) (**Fig 4C and 4D**). Furthermore, receiver operating characteristic curve analysis (ROC) showed that let-7b-5p and *AURKB* expressions may be predictors in BC. The area under the ROC curve for let-7b-5p was 0.893 (95% CI: 0.839–0.946) and for *AURKB* was 0.849 (95% CI: 0.786–0.911) (**Fig 4E and 4F**).

Transfecting let-7b-5p mimics efficiently enhanced let-7b-5p in MCF-7/Dox and MDA-MB-231/Dox cells (P < 0.05) (**Fig 5A**). *AURKB* expression level was significantly reduced in both let-7b-5p mimic transfected cells and barasertib (*AURB* inhibitor) treated cells (P < 0.05) (**Fig 5B and 5C**). Western Blott experiments demonstrated that overexpression of let-7b-5p significantly reduced *AURKB* protein levels in parental and Dox-resistant cells

**Table 4. The correlation scores between putative target genes of let-7b-5p and *AURKB*.**

| GeneName | Description | Corr. | p.Value |
|---|---|---|---|
| *CDCA8* | cell division cycle associated 8 | 0.914 | 1.797e-08 |
| *KIFC1* | kinesin family member C1 | 0.9 | 2.334e-08 |
| *RRM2* | ribonucleotide reductase regulatory subunit M2 | 0.863 | 4.122e-08 |
| *PLK1* | polo like kinase 1 | 0.841 | 5.47e-08 |
| *BIRC5* | baculoviral IAP repeat containing 5 | 0.833 | 6.02e-08 |
| *CCNB2* | cyclin B2 | 0.832 | 6.091e-08 |
| *UHRF1* | ubiquitin like with PHD and ring finger domains 1 | 0.828 | 6.381e-08 |
| *CCNB1* | cyclin B1 | 0.812 | 7.63e-08 |
| *CCNA2* | cyclin A2 | 0.803 | 8.399e-08 |
| *NUSAP1* | nucleolar and spindle associated protein 1 | 0.799 | 8.758e-08 |
| *PTTG1* | pituitary tumor-transforming 1 | 0.796 | 9.034e-08 |
| *AURKA* | aurora kinase A | 0.755 | 1.346e-07 |
| *CKS2* | CDC28 protein kinase regulatory subunit 2 | 0.717 | 1.888e-07 |

Corr: Correlation

(**Fig 5D**). These findings indicated that aberrant expression of let-7b-5p and *AURKB* may be linked to Dox resistance in BC.

## Overexpression of let-7b-5p or depletion of *AURKB* increased doxorubicin sensitivity in breast cancer cells

To determine the effects of let-7b-5p or *AURKB* on the sensitivity of BC cells to Dox, a CCK-8 assay was conducted on MCF-7/Dox and MDA-MB-231/Dox cells transfected with let-7b-5p

**Table 5. Clinicopathological characteristics of the BC patients.**

| Characteristics | Category | Number(%) |
|---|---|---|
| **Age (year)** | <50 | 23(31.9) |
| | >50 | 49(68.1) |
| **Histological grade** | I | 4(5.6) |
| | II | 38(52.8) |
| | III | 30(41.7) |
| **Tumor size** | <2 cm | 16(22.2) |
| | 2–5 cm | 55(76.4) |
| | >5 cm | 1(1.4) |
| **Lymph node status** | Negative | 37(51.4) |
| | Metastasis to ≤3 ln | 31(43.1) |
| | Metastasis to >3 ln | 4(5.6) |
| **Stage** | I | 11(15.3) |
| | II | 56(77.8) |
| | III | 5(6.9) |
| **Molecular subtype** | Luminal A | 30(41.7) |
| | Luminal B | 36(50) |
| | Her2(+) | 1(1.4) |
| | TNBC | 5(6.9) |
| **Ki-67(%)** | <20% | 27(37.5) |
| | ≥20% | 45(62.5) |

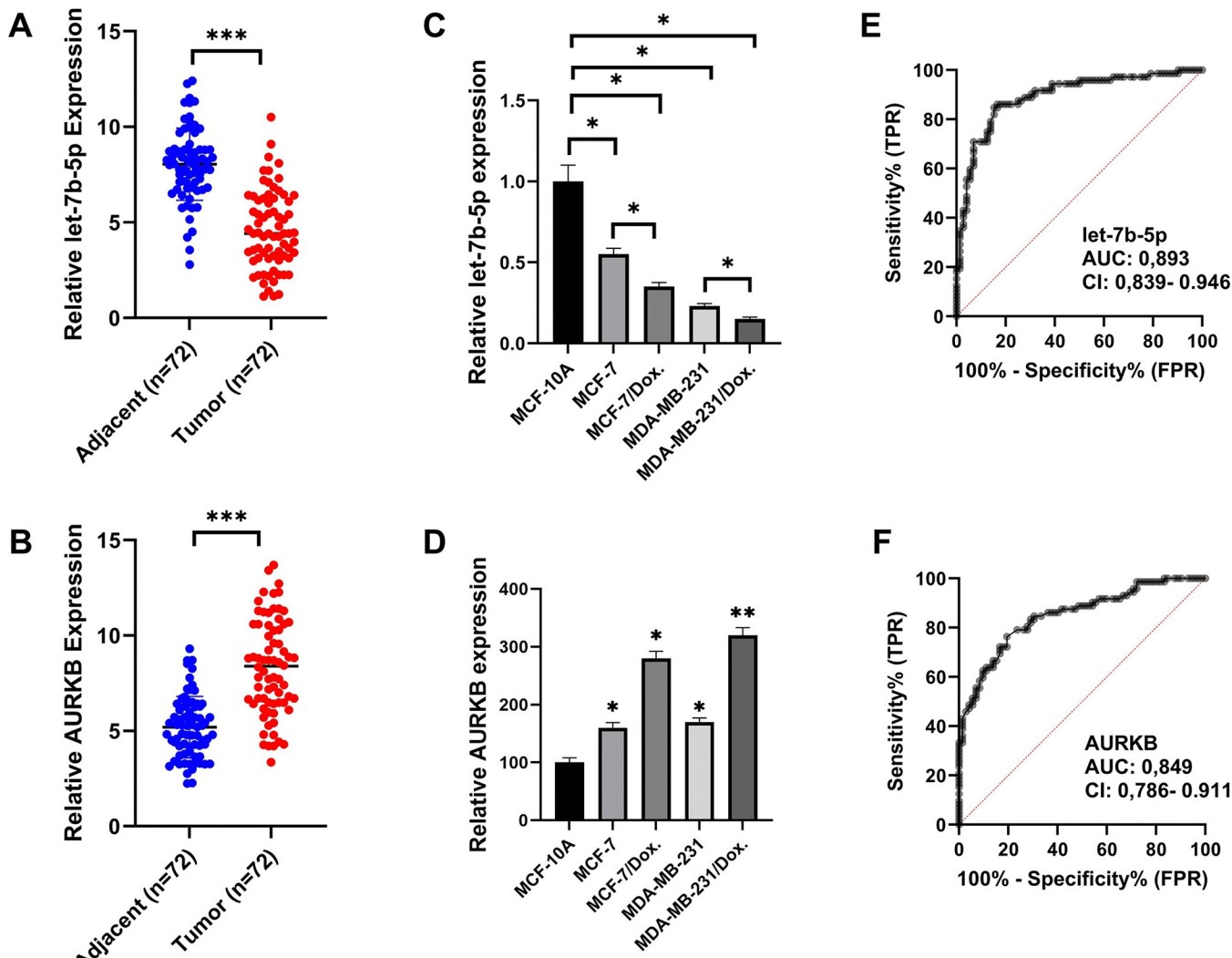

**Fig 4. In BC tissue samples and BC cell lines, the expression of let-7b-5p decreased while the expression of AURKB increased.** Quantification of the relative mRNA expression level of (A) let-7b-5p and (B) AURKB in 72 BC tumor specimens and their matched paracancerous tissue specimens using quantitative PCR. Examination of the (C) let-7b-5p miRNA (D) AURKB mRNA expression levels in different parental and doxorubicin-resistant BC cells. ROC analysis of (E) let-7b-5p and (F) AURKB expressions in BC tissue samples. RNU43 was used as an internal control for miRNA evaluation, and GAPDH was used as an internal control for mRNA evaluation.

mimic or *AURKB* inhibitor. The cells were treated with varying doses of Dox for 48 hours (**Fig 6A and 6B**). The results showed that overexpression of let-7b-5p or downregulation of *AURKB* significantly reduced cell viability at different concentrations of Dox and decreased Dox resistance in MCF-7 and MDA-MB-231 cells compared to the control groups (**Figs 6C–6F and 7**).

Cell migration is a key factor in the occurrence of cancer metastases. A wound healing evaluation assessed the influence of let-7b-5p overexpression and/or *AURKB* inhibition on cell migration. The results indicated that the wound closure rate was considerably diminished in let-7b-5p transfected and/or *AURKB* inhibitor-treated Dox-resistant BC cells compared to the miR-NC or NC groups (**Figs 8 and 9**). Next, we used flow cytometry assay to determine if let-7b-5p-induced Dox sensitivity was associated with apoptosis. The results showed that

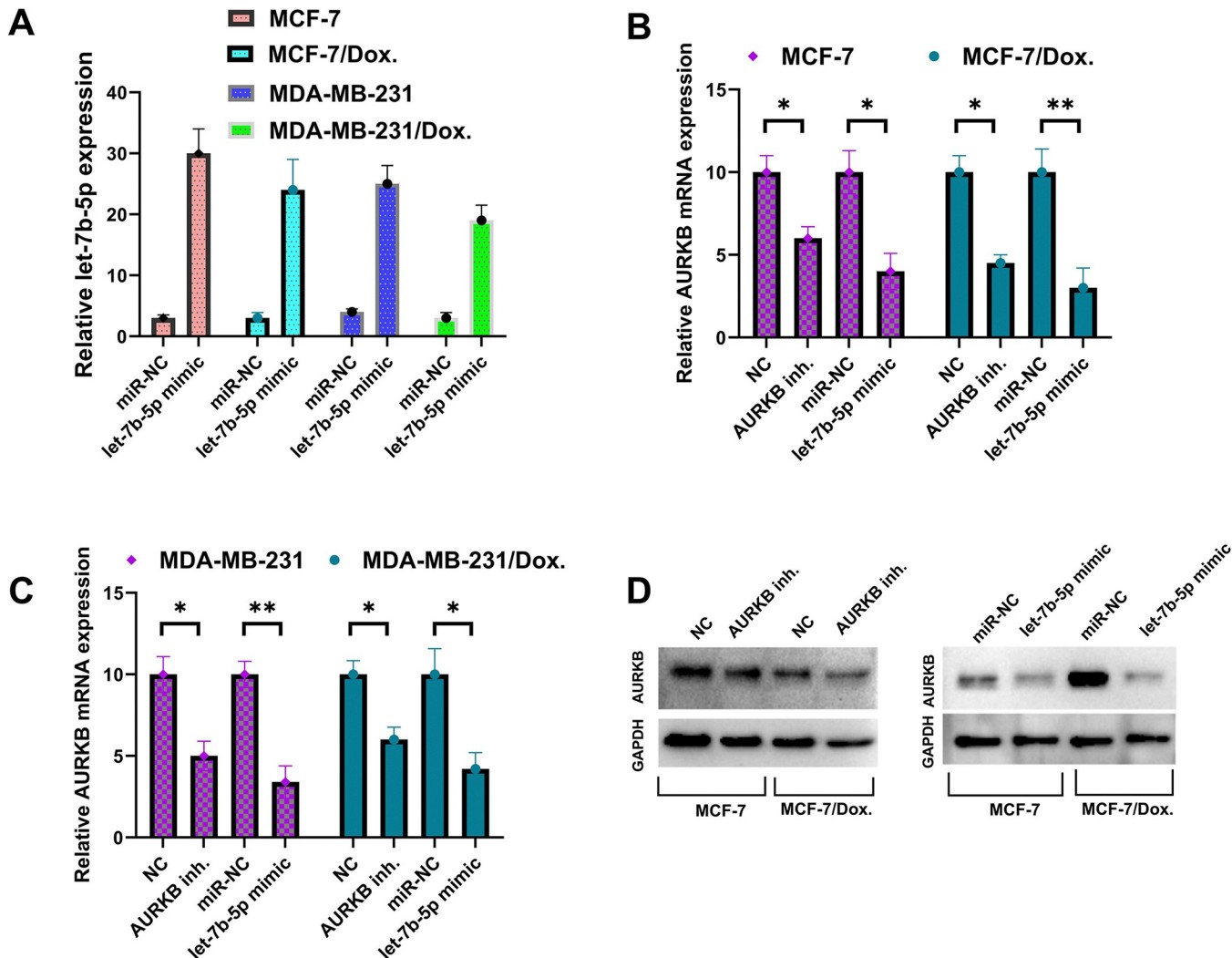

**Fig 5. Restoration of let-7b-5p miRNA expression or inhibition of AURKB expression in parental and doxorubicin-resistant BC cells.** Overexpression of let-7b-5p reduced AURKB expression in BC parental and doxorubicin-resistant cells. (A) The let-7b-5p mimic transfection efficiency was examined by qRT-PCR 24h after transfection. RNU43 was used as an internal control. (B) and (C) Determination of the AURKB mRNA expression levels in parental and doxorubicin-resistant BC cell lines. (D) The protein expression level of AURKB was detected by western blot in cells transfected with let-7b-5p mimics, or miR-NC. GAPDH was used as an internal control in mRNA and protein detection experiments. *P<0.05, **P<0.01, compared with the miR-NC or NC group. MiR-NC: miRNA negative control, NC: Negative control, AURKB inh.: AURKB inhibitor (Barasertib), MCF-7/Dox.: MCF-7 Doxorubicin-resistant cells, MDA-MB-231/Dox.: MDA-MB-231 Doxorubicin-resistant cells.

increased let-7b-5p expression significantly accelerated apoptosis in parental and dox-resistant BC cells (**Fig 10**). Our findings suggested that let-7b-5p overexpression inhibits BC cells and makes BC cells more sensitive to Dox.

## Discussion

Since miRNAs can target numerous genes, miRNA therapy approach can prove to be more advantageous than the single protein target approach [28]. Due to the ability of these compounds to selectively affect many genes within a complex regulatory network, their activities may result in a more extensive and comprehensive therapeutic response [29]. In vitro and in vivo studies have shown that these molecules possess a substantial potential for the cancer

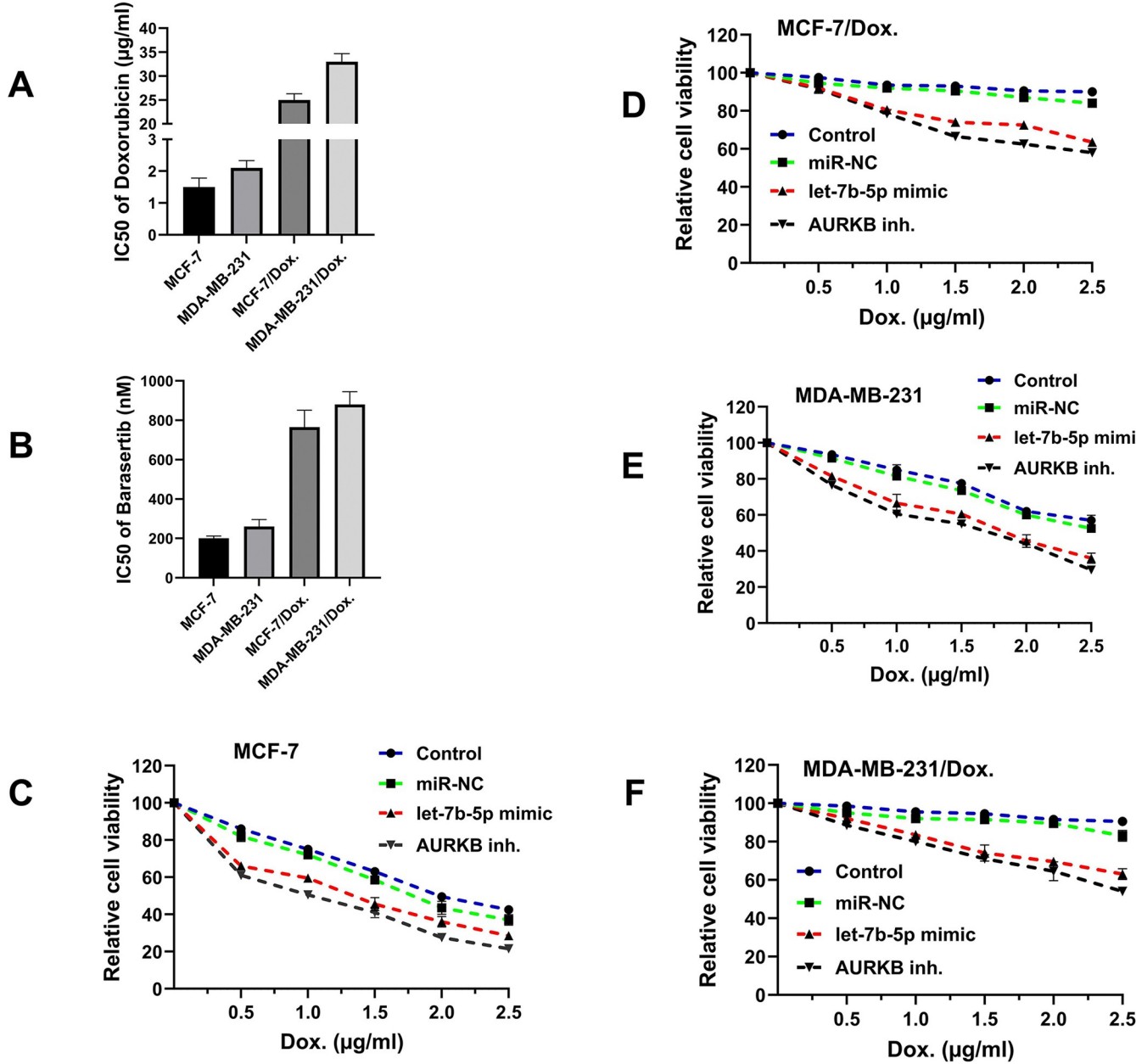

**Fig 6. let-7b-5p restoration or AURKB inhibition sensitized BC cells to doxorubicin.** IC50 values of (A) Doxorubicin and (B) AURKB inhibitor (Barasertib) in parental and doxorubicin-resistant cells. Relative cell viability (mean ± SD) (C) MCF-7, (D) MCF-7/Dox. (Doxorubicin resistant), (D) MDA-MB-231, (E) MDA-MB-231/Dox. (Doxorubicin resistant) cells of let-7b-5p mimic transfected or AURKB inhibitor-treated with doxorubicin for 48 h.

treatment in the future [30]. Although miRNA-based therapeutics have potential, they include risks of immunogenicity and safety concerns. The effectiveness of miRNAs specific to targets may be affected by variables like poor RNA stability, uncertain tumor-specific delivery, and localized effects of miRNAs. Despite the advancement of miRNA-targeting medications in clinical trials, none of them have been registered in the clinicaltrials.gov database for the next phase III [30]. It is imperative to have a more precise understanding of the involvement of miRNAs in molecular pathways to solve this challenge. Hence, it is crucial to elucidate the association between miRNAs and their target genes in different types of cancer to facilitate the

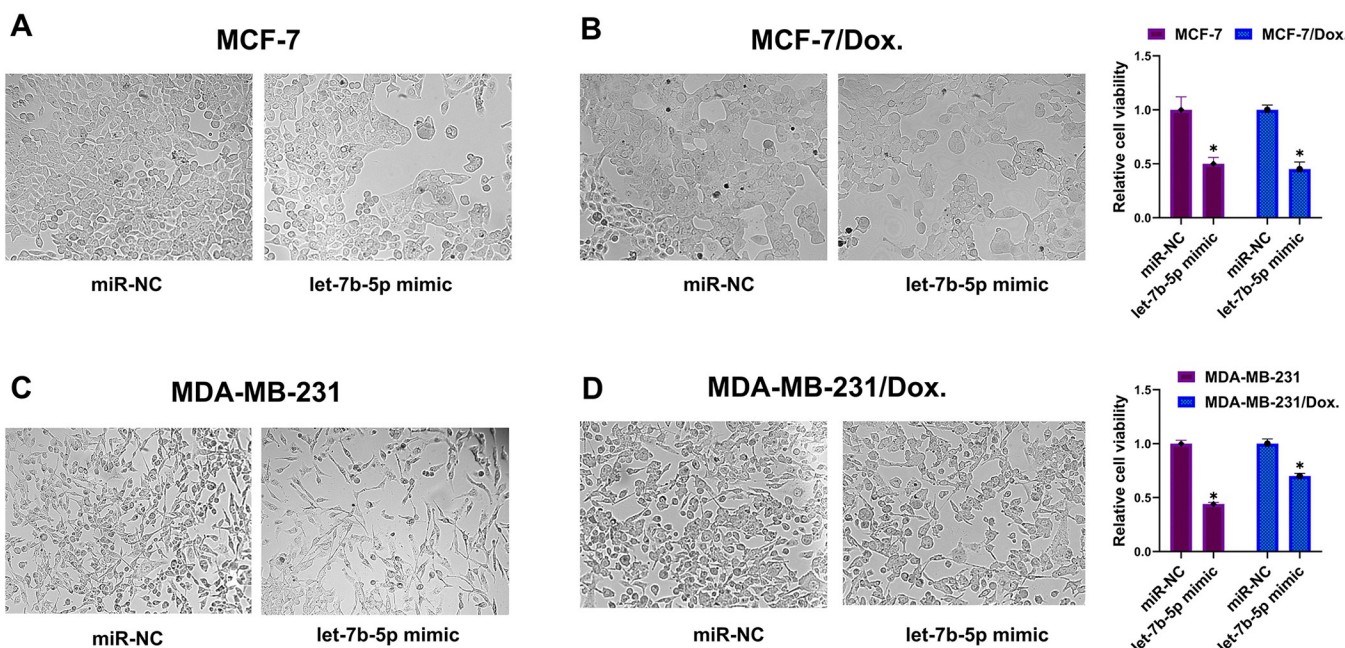

**Fig 7. Cell viability imaging in parental and doxorubicin-resistant cells.** Light microscopy images of (A) MCF-7, (B) MCF-7/Dox, (C) MDA-MB-231, and (D) MDA-MB-231/Dox cells transfected with let-7b-5p or miR-NC after 48 hours. Let-7b-5p overexpression significantly reduced parental and doxorubicin-resistant cell proliferation compared to control groups. The images were captured at 10x magnification. The bars represent mean ± SD, *p < 0.05. Dox.: Doxorubicin resistant.

future utilization of miRNAs as biomarkers and advance the development of therapeutic approaches.

Thousands of potential illness-related miRNAs and genes have been identified using microarray systems and RNA-seq technologies. However, understanding which deregulated genes and/or miRNAs obtained from microarray and RNAseq systems contribute to the disease process is laborious and expensive. Recent advances in bioinformatics methods that facilitate the analysis of microarray and RNA-seq data and the increase in the number of in silico tools developed for the prediction of matches between miRNAs and target genes have made it easier to investigate miRNA-target gene relationships in vitro and in vivo [2].

The examination of GSE45666, GSE154255, and GSE38167 datasets in the bioinformatics section of this work revealed 19 DEmiRNAs in common. After analyzing the BC overall survival relationships of these miRNAs and reviewing the literature, it was determined that the deregulation of let-7b-5p, miR-1290-5p, and let-7c-5p were closely related to BC. When these three miRNAs were searched in Pubmed using the keyword "miRNA name, breast cancer," it turned out that let-7c-5p and miR-1290-5p were connected with BC. Moreover, more information about the association between let-7b-5p and BC was found. Examples to the studies investigating the association between these miRNAs and BC are as follows: Fu et al. demonstrated that let-7c-5p inhibited cell proliferation and promoted cell apoptosis by targeting *ERCC6* in BC [31]. Sun et al. found that let-7c-5p inhibits estrogen-activated Wnt signaling in BC stem cell self-renewal stimulation [32]. miR-1290-5p controls the radioresistance of triple-negative BC via inhibiting *NLRP3*-mediated pyroptosis [33]. miR-1290-5p triggers astrocytes in the brain metastatic microenvironment through the *FOXA2*/*CNTF* axis, promoting the spread of brain metastases [34]. Wang et al. demonstrated that overexpressing *CDX2* reduced BC by upregulating let-7b-5p and decreasing *COL11A1* expression [35]. It was demonstrated

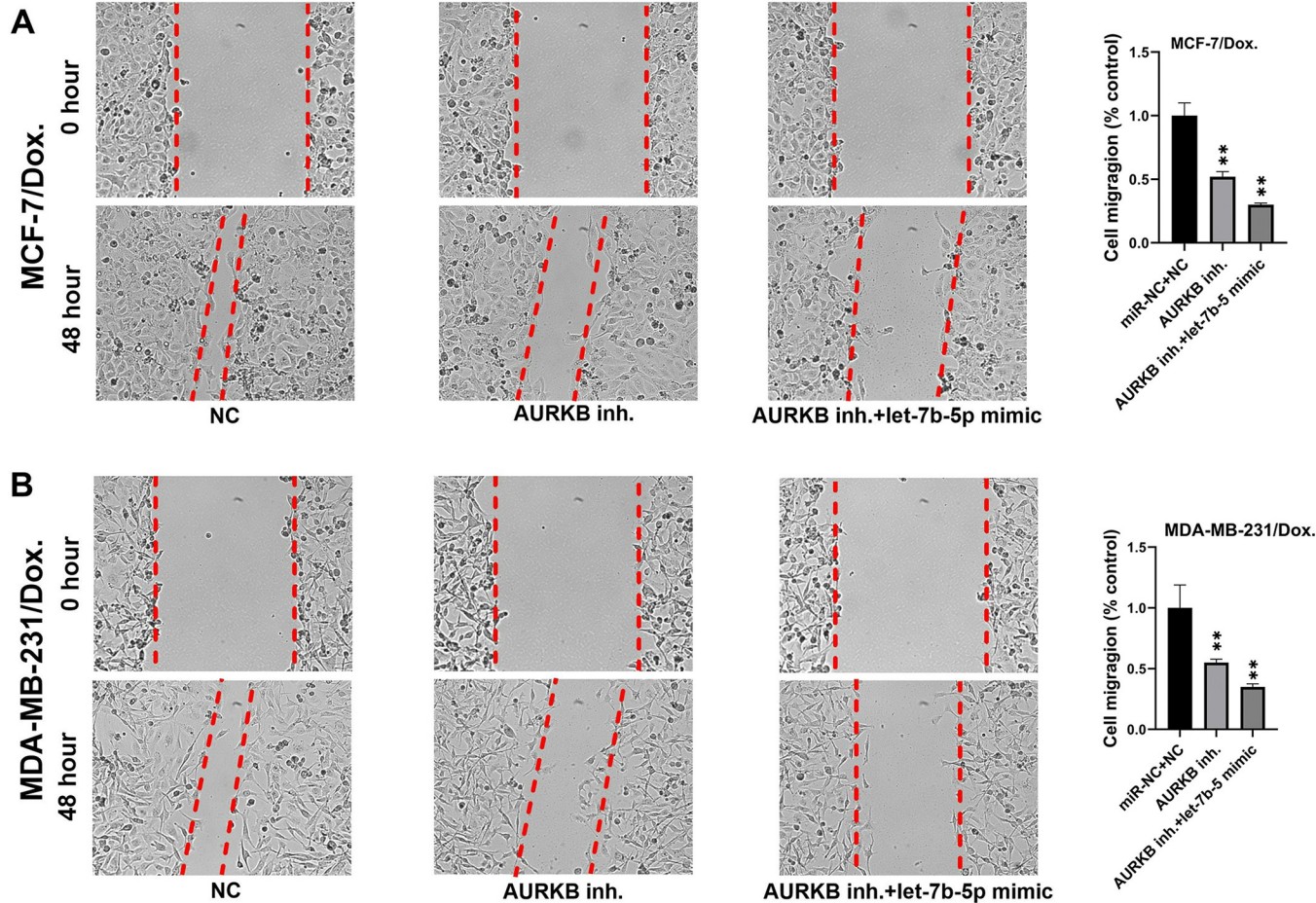

**Fig 8.** The AURKB inhibitor (Barasertib) reduced migration in (A) doxorubicin-resistant MCF-7 cells and (B) MDA-MB-231 doxorubicin-resistant cells. Furthermore, migration was dramatically reduced in doxorubicin-resistant cells with the combination of transfection of a let-7b-5p mimic and AURKB inhibitor treatment.

that let-7b-5p suppressed the cancer-promoting functions of BC-associated fibroblasts via IL-8 inhibition [36]. These findings indicate that let-7b-5p, miR-1290-5p, and let-7c-5p may play a role in the progression of BC. Although let-7b-5p was chosen for in vitro research in our study, further research on let-7c-5p and miR-1290-5p is recommended since they may have roles in drug resistance and other BC pathways.

Doxorubicin is one of the most widely used drugs in the treatment of BC. However, its effectiveness is not at the desired level due to the drug resistance. let-7b-5p downregulation has been demonstrated in many cancers, including BC [37,38]. Let-7b-5p has been shown to play a role in drug resistance mechanisms in various cancers. For instance, it was shown that the polymorphism in the 3'UTR binding region of the *BCL-2* gene, targeted by let-7b, in B-cell lymphoma was associated with fluorouracil resistance of hepatocellular carcinoma [39]. Few studies have shown that let-7b-5p may play a role in Dox resistance in cancers. For example, A single nucleotide polymorphism (SNP) in the let-7b binding in the *Bcl-xL* gene's 3'-UTR site has been found to promote resistance to 5-fluorouracil and Dox in BC cells [40]. However, the relationship between let-7b-5p and Dox resistance in BC has not yet been identified.

Previous studies have demonstrated that let-7b-5p directly targets *AURKB*, leading to a reduction in the expression of this gene at both mRNA and protein levels in different diseases.

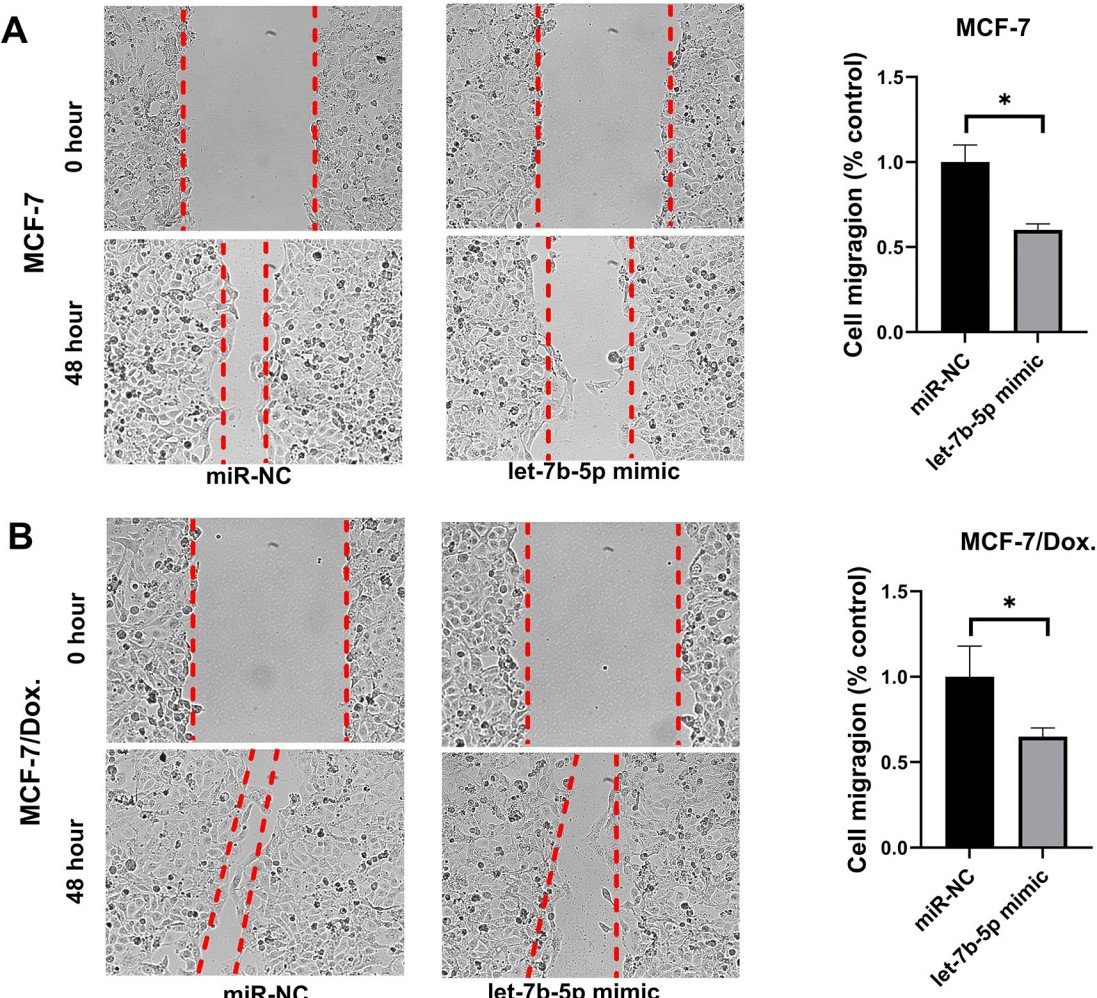

**Fig 9. Wound healing assay on the cell migration capability of parental and doxorubicin-resistant BC cells following transfection of let-7b-5p mimic.** Evaluation of the cell migration distances in (A) MCF-7 (B) MCF-7/Dox. cells. Let-7b-5p overexpression in both the parental and doxorubicin-resistant cells significantly reduced migration compared to the control groups.

For instance, Han et al. demonstrated that let-7b-5p reduced cisplatin resistance and tumor development by blocking *AURKB* in gastric cancer [26]. In another study, Zhou et al. demonstrated that let-7b-5p regulates adriamycin (doxorubicin) resistance via *AURKB* in chronic myeloid leukemia cells [27]. These findings from the literature demonstrate an important link between let-7b-5p, *AURKB*, and drug resistance and support our findings.

The *AURKB* gene is overexpressed in many cancer types including BC [41–43]. Pellizzari et al reported that inhibition of the *AURKB* pathway may be a promising strategy for the treatment and radiosensitization of Triple Negative Breast Cancer (TNBC) [44]. It has been demonstrated that *AURKB* stimulates epithelial-mesenchymal transition by stabilizing Snail1, resulting in basal-like breast cancer metastases [45]. *AURKB* is also closely associated with drug resistance. For instance, Liu et al., revealed that *AURKB* is related to paclitaxel resistance in BC cells through the *PRKCE/RAB27B* axis, suggesting a possible intervention target to reverse tumor drug resistance [46]. A different investigation has demonstrated that *AURKB* facilitates the development of tumors and carboplatin resistance via controlling the *ERK* pathway in neuroblastoma cells [47].

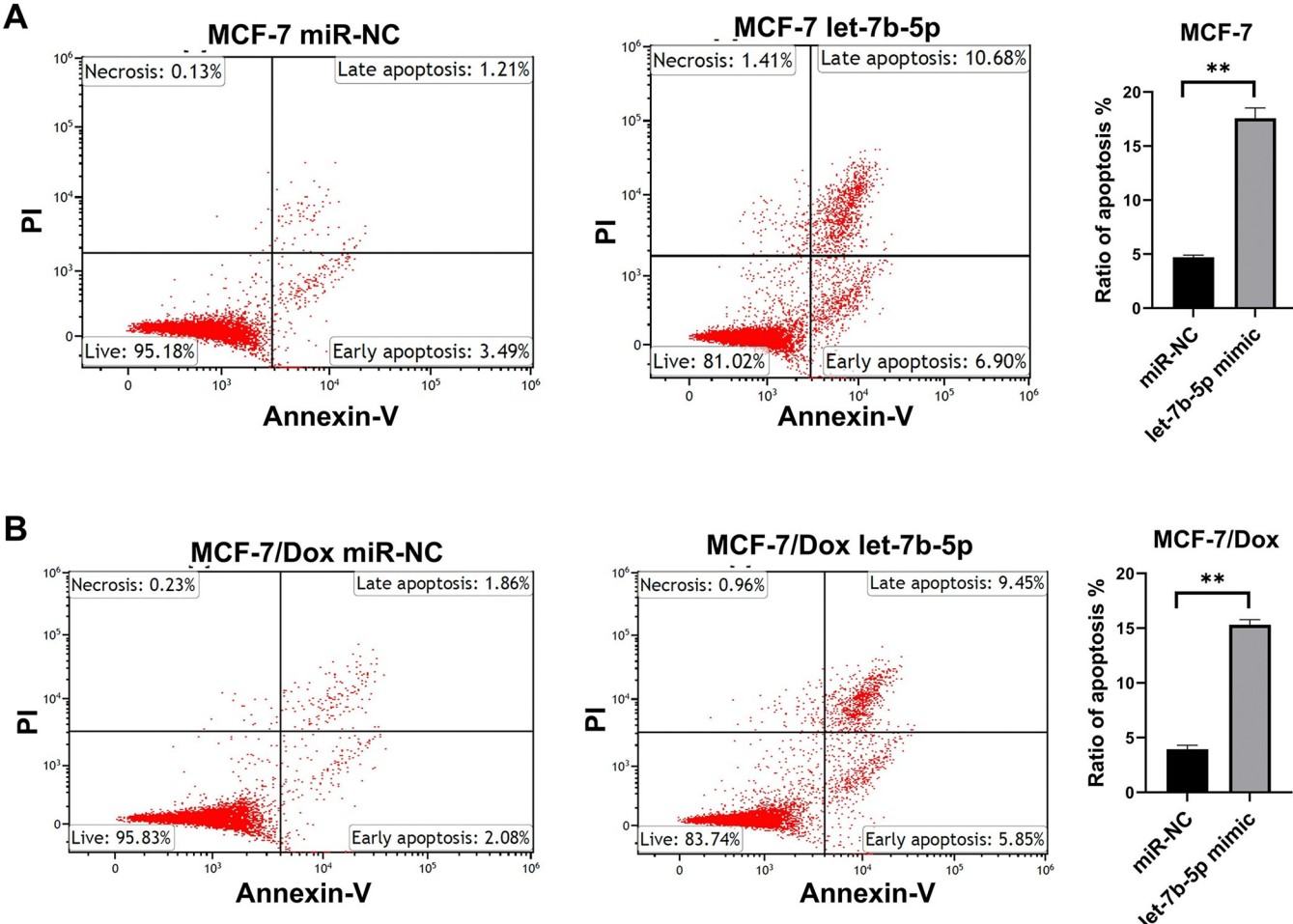

**Fig 10. The effect of let-7b-5p mimic transfection on apoptosis of BC cells.** Overexpression of let-7b-5p promoted apoptosis of (A) MCF-7 and (B) MCF-7/Dox cells **; $P < 0.01$.

Some studies have also shown that miRNAs can specifically target *AURKB* and are associated with drug resistance in cancers. For example, miR-486-5p has been shown to regulate DNA damage inhibition and cisplatin resistance in lung adenocarcinoma via AURKB [48]. The results of these studies indicate that *AURKB* may be responsible for drug resistance in certain malignancies through various molecular mechanisms, such as miRNAs. The provided research supports that let-7b-5p may be linked to Dox resistance in BC through *AURKB*.

In current study a comparison of the possible target genes of let-7b-5p in the miRNet tool between TCGA-BC data, GSE100925, and GSE229571 BC GEO datasets revealed 14 genes in common. The STRING analysis revealed a strong link between these 14 genes. Furthermore, a significant association was identified between *AURKB* and the other 13 genes. These genes have been shown to possess potential oncogenic roles in several malignancies, including BC. For instance, Huang et al. showed that *CKS2* overexpression is a poor prognostic marker and promotes cell proliferation and invasion in BC [49]. Overexpression of *CKS2* has been shown to enhance the effectiveness of chemotherapy by overriding DNA damage regulators [50]. It has been determined that upregulation of *NUSAP1* expression stimulates cell proliferation in invasive BC cells via *CDK1* and *DLGAP5* and is related to Epirubicin resistance [51]. *CCNA2* serves as a possible indicator for immunotherapy in BC and acts as a prognostic indicator for

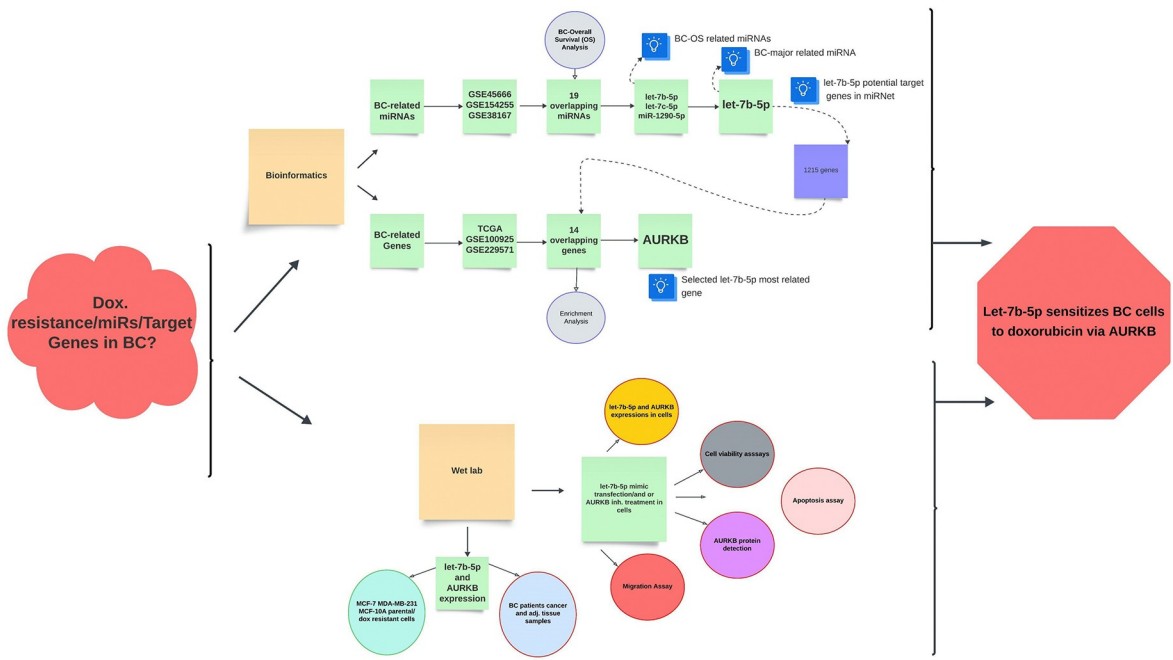

**Fig 11. Graphical abstract of the study.** The relationship between miRNAs and target genes in BC doxorubicin resistance was investigated following bioinformatics and wet lab study. In the bioinformatics part publicly available geo datasets were used for DEmiRNAs and DEMs selection. We have identified 19 shared DEmiRNAs in the 3 geo datasets. OS analysis was performed with the Kmplot tool, and it was found that let-7b-5p, miR-1290-5p, and let-7c-5p were significant in BC in both TCGA and METABIRIC data. According to Pubmed, let-7b-5p was chosen for wet lab research because it contained more results about BC associations than the other two miRNAs. DEMs were selected using TCGA-BC, GSE100925 and GSE229571 geo datasets. 14 shared genes (let-7b-5p potential targets) were found. According to the literature review and correlation analysis results, the AURKB gene was chosen as a candidate for wet lab research because it was seen to be more closely related to let-7b-5p among the overlapping genes. In the wet lab part of the study, let-7b-5p and AURKB expression levels were investigated in breast cancer tissue samples and cell lines (parental and/or doxorubicin-resistant). The effect of the let-7b-5p/AURKB axis on cellular processes was investigated by transfecting let-7b-5p mimic parental cells and doxorubicin-resistant cells and/or treating AURKB inhibitor (Barasertib). As a result, our findings showed that let-7b-5p sensitized breast cancer cells to doxorubicin resistance through the AURKB gene.

ER+ BC and tamoxifen resistance [52,53]. *RRM2* is another prognostic indicator in BC and is associated with worse survival outcomes and tamoxifen resistance [54]. *RRM2* silencing with siRNA has been shown to significantly inhibit pancreatic tumor proliferation, either alone or with Dox [55]. *CDCA8* is another prominent gene that may be related to *AURKB*. It is a key modulator of estrogen-induced cell proliferation in BC cells [56]. The *AURKB/CDCA8* pathway is potentially pivotal in cancer pathogenesis. Hayama et al. showed that the phosphorylation and activation of *CDCA8* by *AURKB* contribute to human lung carcinogenesis [57]. *PLK1*, whose increased expression is a major problem in BC, has been reported to mediate resistance to Palbociclib in HR+/HER2- metastatic BC [58]. According to studies, *BIRC5* (survivin), an inhibitor of apoptosis (IAP) family member was substantially expressed in BC patients and, it also promotes resistance to chemotherapy, anti-HER2 treatment, and radiotherapy [59]. Mitotic kinases *PLK1* and *AURKB* control BC cell proliferation by phosphorylating *BIRC5* [60]. The literature that was assembled here suggests that the molecular linkage of genes connected to *AURKB* should be examined in more depth. This would allow for a better understanding of the processes that cause drug resistance in BC, as well as the development of novel treatments. Based on these findings, it has been suggested that the let-7b-5p/*AURKB* axis may affect cancer processes by influencing several key genes.

GEO2R is a web-based tool that is user-friendly and enables users to analyze various data sets from a GEO series in order to identify genes, miRNAs, circRNAs, and other molecules that are differentially expressed. Similarly, miRNet [20] is a web-based platform that is user-friendly and has been specifically designed to assist in the elucidation of miRNA effects by integrating users' data with existing knowledge through network-based visual analytics. Nevertheless, it is important to keep in mind that the results obtained using GEO2R, miRNET, and other in silico tools in the study may contain data that could restrict the study. For instance, in the GEO2R analysis, some miRNAs, excluded based on logFC and p-value, may have significant relevance in elucidating the link between BC and Dox resistance. In our work, findings based on miRTarbase and Tarbase tools derived from the miRNET tool used to identify the possible target genes of the chosen miRNA (let-7b-5p). For the prediction of miRNA-Gene interactions, other target genes might be proposed for let-7b-5p in tools other than miRTarbase and Tarbase (e.g. miRDB). Thus, it is important to note that, in addition to the miRNAs and genes identified through various GEO datasets and in silico tools, certain significant miRNAs and genes may have been excluded due to the limitations of the bioinformatics methods employed. The investigation provided significant data regarding the relationship between BC Dox resistance/let-7b-5p/*AURKB*; however, the lack of in vivo experiments represents a different limitation of the study.

In this study, using bioinformatics methods, we determined that let-7b-5p and *AURKB* may be essential molecules in BC. In silico analysis revealed that *AURKB* is one of the strongest candidates among the probable targets of let-7b-5p and they exhibit a negative correlation in BC. Thus, we examined let-7b-5p and *AURKB* expressions in BC and adjacent tissue samples. In accordance with the literature, let-7b-5p was significantly downregulated, and *AURKB* was upregulated in BC tissue samples. Then, our investigation was conducted into the intricate relationship between let-7b-5p and *AURKB* and their functional effects on parental BC and Dox-resistant cells. According to our in vitro-study findings, the restoration of let-7b-5p is a potential inhibitor of parental and Dox-resistant cell proliferation, triggering apoptosis and reducing migratory ability. The significant decrease of *AURKB* at both mRNA and protein levels in let-7b-5p transfected parental and Dox-resistant cells was considered as an important sign that let-7b-5p may function through *AURKB* in BC cells (**Fig 11**).

In conclusion, our findings imply that the let-7b-5p/*AURKB*/drug resistance link may be in BC. Therefore, it may be vital to conduct deeper investigations on this topic in relation to BC.

## Acknowledgments

We thank Prof. Dr. Neslihan Abaci from Istanbul University Aziz Sancar Institute of Experimental Medicine Genetics Department and Prof. Dr. Mehmet Topcul Istanbul University, Faculty of Sciences, Department of Biology, for providing us with cell lines.

## Author Contributions

**Conceptualization:** Zeyneb Kurt.

**Data curation:** Murat Kaya, Semen Önder.

**Formal analysis:** Murat Kaya.

**Funding acquisition:** Murat Kaya.

**Investigation:** Murat Kaya, Ilknur Suer, Semen Önder, Sukru Palanduz.

**Methodology:** Murat Kaya, Asmaa Abuaisha, Ilknur Suer, Evren Onay Ucar, Mustafa Nuri Yenerel, Kivanc Cefle, Zeyneb Kurt.

**Project administration:** Selman Emiroglu, Sukru Ozturk.

**Resources:** Selman Emiroglu.

**Supervision:** Sukru Ozturk, Zeyneb Kurt.

**Validation:** Murat Kaya, Asmaa Abuaisha, Ilknur Suer, Evren Onay Ucar.

**Visualization:** Murat Kaya.

**Writing – original draft:** Murat Kaya, Asmaa Abuaisha.

**Writing – review & editing:** Ilknur Suer, Selman Emiroglu, Semen Önder, Evren Onay Ucar, Mustafa Nuri Yenerel, Sukru Palanduz, Kivanc Cefle, Sukru Ozturk, Zeyneb Kurt.

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
