## [Decision Letter · Decision Letter 0]

23 Aug 2024

PONE-D-24-25735Let-7b-5p sensitizes breast cancer cells to doxorubicin through Aurora Kinase BPLOS ONE

Dear Dr. Kurt,

Thank you for submitting your manuscript to PLOS ONE. After careful consideration, we feel that it has merit but does not fully meet PLOS ONE’s publication criteria as it currently stands. Therefore, we invite you to submit a revised version of the manuscript that addresses the points raised during the review process.

We look forward to receiving your revised manuscript.

Kind regards,

Ruo Wang

Academic Editor

PLOS ONE

 [This research was supported by the Council of Higher Education Research Universities Support Program project, ADEP-Istanbul University, TSA-2023-39483.].  

Additional Editor Comments:

The reviewers provided very detailed suggestions for revisions. Many of them coincide with my views. The more important ones are as follows:

First, the manuscript relies to some extent on in silicon lab methods to advance, and it should consider adding in vivo experimental data to improve the limitations of this part of the method (or at least, explain it).

The manuscript should consider discussing genes related to the AURKB gene. Or better yet, the relationship between this part of the genes and doxorubicin resistance.

The manuscript should consider answering the relationship between AURKB and breast cancer subtypes or other tumors.

In summary, the manuscript should be considered for acceptance after revision.

Reviewers' comments:

Reviewer's Responses to Questions

**Comments to the Author**

1. Is the manuscript technically sound, and do the data support the conclusions?

Reviewer #1: Yes

Reviewer #2: Yes

Reviewer #3: Yes

2. Has the statistical analysis been performed appropriately and rigorously? 

Reviewer #1: Yes

Reviewer #2: Yes

Reviewer #3: I Don't Know

3. Have the authors made all data underlying the findings in their manuscript fully available?

Reviewer #1: Yes

Reviewer #2: Yes

Reviewer #3: Yes

4. Is the manuscript presented in an intelligible fashion and written in standard English?

Reviewer #1: Yes

Reviewer #2: Yes

Reviewer #3: Yes

5. Review Comments to the Author

Reviewer #1: The manuscript titled "Let-7b-5p sensitizes breast cancer cells to doxorubicin through Aurora Kinase B" presents an interesting exploration of the relationship between miRNA let-7b-5p, Aurora Kinase B (AURKB), and doxorubicin resistance in breast cancer. While the study touches on a crucial aspect of breast cancer therapy and drug resistance, several significant concerns need to be addressed to strengthen the overall quality and impact of the research.

o The study relies heavily on bioinformatics analyses, which is a valid approach. However, the validation of in silico findings should be more robust, especially considering the clinical implications of the results. The current data from quantitative PCR and cell line studies, while promising, seem insufficient to conclusively establish the role of let-7b-5p/AURKB in doxorubicin resistance. Additional experiments, such as in vivo models or patient-derived xenografts, would provide stronger evidence.

o The Discussion could benefit from a more focused discussion that directly relates previous studies to the findings of this research. For instance, while other studies have linked let-7b-5p to drug resistance in different cancers, the manuscript should clearly distinguish how this study’s findings on AURKB add to or differ from the existing knowledge.

o The mechanistic pathway connecting let-7b-5p and AURKB to doxorubicin resistance is not thoroughly explored. The manuscript would be significantly strengthened by elucidating how this miRNA-target interaction specifically influences cellular pathways leading to drug resistance. Details on the downstream effects of AURKB inhibition by let-7b-5p, such as alterations in apoptosis or cell cycle regulation, should be provided.

o The results would benefit from a more detailed and systematic presentation of the data. For example, the relationship between let-7b-5p and AURKB in both parental and doxorubicin-resistant cell lines should be explicitly compared and contrasted. Additionally, the inclusion of statistical analyses for all experiments would help clarify the significance of the findings.

o The conclusion posits that the let-7b-5p/AURKB axis is potentially relevant across multiple cancers. However, this claim would be more convincing if supported by data from additional cancer types or a broader range of breast cancer subtypes. Future directions should be clearly outlined, focusing on how these findings could translate into clinical applications, such as miRNA-based therapies.

Reviewer #2: The study addresses an important clinical challenge in breast cancer treatment: drug resistance, particularly resistance to doxorubicin, a commonly used chemotherapy drug. The exploration of the let-7b-5p/AURKB axis presents a novel target that could enhance the effectiveness of chemotherapy, potentially leading to better patient outcomes; however, there are some points that needs to be considered before publication:

1. While the study focuses on the let-7b-5p/AURKB axis, it does not extensively explore other miRNAs or genes that may also play critical roles in doxorubicin resistance, which could limit the comprehensiveness of the findings.

2. The manuscript suggests potential clinical applications but lacks in-depth discussion on how these findings could be translated into clinical settings or what challenges might be encountered.

3. The reliance on bioinformatics tools while powerful may introduce biases or limitations inherent to the datasets or tools used such as GEO2R and miRNet. There should be a discussion on potential limitations or biases these tools could introduce.

4. The study does not include in vivo validation which is crucial to understand how the findings would translate to a whole organism particularly in a clinical context.

5. The manuscript would benefit from a more detailed exploration of the functional implications of let-7b-5p/AURKB axis beyond doxorubicin resistance possibly examining other chemotherapeutic agents or combination therapies.

6. The discussion on the STRING analysis and the role of overlapping genes could be expanded to provide a deeper understanding of how these genes interact and contribute to cancer progression.

7. Some sections particularly the introduction and discussion could benefit from more scientific language and a stronger focus on the most critical findings. Redundant detailed explanations of background information should be minimized.

8. There is a need for a more explicit linkage between the study’s findings and the potential clinical applications providing a clearer pathway for future research.

9. The language of overall manuscript needs to be more scientific and grammatical errors needs to be checked thoroughly.

10. All the images should be revised in higher resolution of 600 dpi as the quality is very poor.

Reviewer #3: Here, in this work authors identified target for miRNA let-7b-5p, and used AURKB as a target to interpret the drug resistance. Some comments related to are

1. Include a figure to represent the steps used in the identification of differentially expressed MiRNA.

2. What actually logFC calculates?

3. Check italics case and abbreviation.

4. Discuss the other genes that are in connection/interaction with AURKB gene.

5. Quality of figure 11 is very poor and blurred. Please provide a good quality figures.

6. Is AURAKB target for Breast cancer therapy only ? or for other types cancers also.

6. PLOS authors have the option to publish the peer review history of their article (what does this mean?). If published, this will include your full peer review and any attached files.

Reviewer #1: **Yes: **Abolfazl Bahrami

Reviewer #2: **Yes: **Dr Shama Khan

Reviewer #3: **Yes: **Dev Bukhsh Singh

---

## [Author Response · Author response to Decision Letter 0]

23 Oct 2024

Dear Editor,

We thank the reviewers for their constructive and thoughtful comments and suggestions on our manuscript. 

We have provided our point-to-point responses to the reviewers’ comments and uploaded them within a document named "plos one response to reviewers_ZK.docx".

The modifications in the manuscript are highlighted within the “Manuscript revised 07.10.2024_ZK.docx” document. We also provided an unmarked version of the revised manuscript “Manuscript 07.10.2024_Unmarked_ZK.docx” upon your request.

Sincerely,

Dr Zeyneb Kurt

---

## [Editor Report · Decision Letter 1]

31 Oct 2024

Let-7b-5p sensitizes breast cancer cells to doxorubicin through Aurora Kinase B

PONE-D-24-25735R1

Dear Dr. Kurt,

We’re pleased to inform you that your manuscript has been judged scientifically suitable for publication and will be formally accepted for publication once it meets all outstanding technical requirements.

Kind regards,

Ruo Wang

Academic Editor

PLOS ONE
---

## [Editor Report · Acceptance letter]

12 Nov 2024

PONE-D-24-25735R1 

PLOS ONE

Dear Dr. Kurt, 

I'm pleased to inform you that your manuscript has been deemed suitable for publication in PLOS ONE. Congratulations! Your manuscript is now being handed over to our production team.

Kind regards, 

on behalf of

Dr. Ruo Wang 

Academic Editor

PLOS ONE